# Diversity in kinetics correlated with structure in nano body-stabilized LacY

**Hemant Kumar[1]¤, Janet Finer-Moore[1], Irina Smirnova[2], Vladimir Kasho[2], Els Pardon[3,4], Jan Steyaert [3,4], H. Ronald Kaback[2,5,6]†, Robert M. Stroud [1]***

1 Department of Biochemistry and Biophysics, University of California, San Francisco, California, United States of America, 2 Department of Physiology, University of California, Los Angeles, California, United States of America, 3 VIB-VUB Center for Structural Biology, VIB, Brussels, Belgium, 4 Structural Biology Brussels, Vrije Universiteit Brussels, Brussels, Belgium, 5 Department of Microbiology, Immunology, and Molecular Genetics, University of California, Los Angeles, California, United States of America, 6 Molecular Biology Institute, University of California, Los Angeles, California, United States of America

† Deceased.
¤ Current address: Molecular Biology Unit, Institute of Medical Science, Banaras Hindu University, Varanasi, India.
* stroud@msg.ucsf.edu

## Abstract

The structure of lactose permease, stabilized in a periplasmic open conformation by two Gly to Trp replacements (LacYww) and complexed with a nanobody directed against this conformation, provides the highest resolution structure of the symporter. The nanobody binds in a different manner than two other nanobodies made against the same mutant, which also bind to the same general region on the periplasmic side. This region of the protein may represent an immune hotspot. The CDR3 loop of the nanobody is held by hydrogen bonds in a conformation that partially blocks access to the substrate-binding site. As a result, $k_{on}$ and $k_{off}$ for galactoside binding to either LacY or the double mutant complexed with the nanobody are lower than for the other two LacY/nanobody complexes though the $K_d$ values are similar, reflecting the fact that the nanobodies rigidify structures along the pathway. While the wild-type LacY/nanobody complex clearly stabilizes a similar 'extracellular open' conformation in solution, judged by binding kinetics, the complex with wild-type LacY did not yet crystallize, suggesting the nanobody does not bind strongly enough to shift the equilibrium to stabilize a periplasmic side-open conformation suitable for crystallization. However, the similarity of the galactoside binding kinetics for the nanobody-bound complexes with wild type LacY and with LacY$_{WW}$ indicates that they have similar structures, showing that the reported co-structures reliably show nanobody interactions with LacY.

## Introduction

The lactose permease of *Escherichia coli* (LacY) catalyzes β-galactoside/H⁺ symport across the membrane, using an alternating access mechanism[1]. LacY is the most extensively studied symporter in the Major Facilitator Superfamily and a paradigm for defining transport mechanisms. The protein consists of two 6-helix bundles (an N- and a C-terminal domain)

**Data Availability Statement:** The structure factors and coordinates are available from the Protein Data Bank (www.pdb.org) with accession code 6VBG.

**Funding:** NIH R01 GM024485 (RMS) www.NIH. gov, NIH R01 GM120043 (HRK) www.NIH.gov, NSF Eager Grant MCB1747705 (HRK) INSTRUCT-ERIC (JS) instruct-eMRric.eu, Research Foundation Flanders (JS) www.fwo.be, Program for Breakthrough Biomedical Research, Sandler Foundation (RMS) https://pbbr.ucsf.edu/, MR-15-328559 University of California Office of the President, Multicampus Research Programs and Initiatives (RMS) www.ucop.edu. The funders had no role in study design, data collection and analysis, decision to publish or preparation of the manuscript.

**Competing interests:** The authors have declared that no competing interests exist.

connected by a relatively long cytoplasmic loop[2]. LacY alternates between inward (cyto-plasmic)-open and outward (periplasmic)-open conformations, each of which can bind or release cargo, and according to the kinetic scheme, multiple conformers are involved in the overall transport cycle[1].

Crystal structures have been determined for LacY in an inward-open conformation[2–5] and also in a partially occluded outward-open conformation generated by using two mutations of glycines to tryptophans, mutants G46W/G262W (LacY$_{WW}$)[6] with bound lactose analogs [7, 8]. LacY is highly dynamic and transitions through several additional intermediate confor-mations during turnover[9–13]. We aim to trap some of these conformations by using single-domain camelid nanobodies (Nbs) and to determine their structures by X-ray crystallography in order to define a more complete stereochemical mechanism of symport. The small size of the Nbs and the flexibility of the Complementarity Determining Regions (CDRs) makes them useful tools for stabilizing different conformational states of flexible proteins[14, 15].

In order to stabilize the periplasmic-open conformation of LacY, we generated Nbs against the double-Trp mutant LacY$_{WW}$ in which two introduced Trp residues fall between the N- and C-terminal domains on the periplasmic side and constrain the protein to a periplasmic-open conformation[6]. Nbs developed against outward-open LacY$_{WW}$ typically bind with ~nano Molar affinity to the periplasmic surface with 1:1 stoichiometry (18, 20). Binding of Nb blocks H$^{+}$/galactoside symport catalyzed by WT LacY, but increases the rate of sugar binding by 5 to 50-fold[16, 17]. Moreover, distance-dependent fluorescence quenching/unquenching studies show different extents of opening of the periplasmic side and closing of the cytoplasmic cavity suggesting that the Nbs selectively trap different transient conformers of LacY[18].

Crystal structures of two complexes of Nbs with LacY$_{WW}$ have been determined[19, 20] demonstrating different conformations, as predicted from fluorescence experiments[18]. Nb9039/apo-LacY$_{WW}$ exhibits a more open periplasmic cavity than Nb9047/LacY$_{WW}$ with bound *p*-nitrophenyl-α-D-galactopyranoside (NPG). Here we report the structure of the Nb9043/LacY$_{WW}$ complex with bound β-D-galactopyranosyl-1-thio-β-D-galactopyranoside (TDG). The conformation of LacY$_{WW}$ is similar to that of Nb9047/LacY$_{WW}$ with bound NPG although Nb9043 has a much smaller Complementarity Determining Region 3 (CDR3) than Nb9047 and binds in a shifted position. As proposed for the previously described complexes, a hydrogen-bond network that bridges the N- and C-terminal domains via CDR3 may deter-mine how the outward-open conformation is stabilized.

## Results

### Overall structure

LacY$_{WW}$ was co-crystallized in complex with Nb9043 by vapor diffusion in the presence of TDG. The crystals were in space group P65 and contain two complexes of LacY$_{WW}$/TDG/Nb9043 in the asymmetric unit with unit cell axes a = 151.3Å and c = 182.6Å. The crystals dif-fracted to 2.8Å resolution, and the structure was solved by molecular replacement using the previous LacY$_{WW}$ structure with bound Nb9047 and NPG (PDB ID 6C9W) [20], but with NPG and Nb removed from the search model. The protein structure was refined against the X-ray data to a free R-factor of 27.5% (PDB ID 6VBG).

There are two approximately two-fold related LacY$_{WW}$ /Nb complexes in the asymmetric unit. We label the two independent LacY$_{WW}$ chains A and B, with attached Nb9043s labeled C and D, respectively. Both complexes have the same structure and are closely packed, with a 566Å$^{2}$ interface comprising 13 residues from each LacY$_{WW}$ chain. This arrangement isn't physiological since it would orient LacY in opposite directions with respect to the membrane. Crystal contacts consist of two types of interfaces involving the Nb9043 molecules. An Nb

chain from one of the complexes packs against the Nb chain from the second complex that is related by crystallographic symmetry (C-D' interface), and also packs against LacY from the same complex related by a different crystallographic symmetry operation (C-A" or D-B" interface) (S1 Fig).

Residues 190–202 of LacY$_{WW}$, comprising most of the long central cytoplasmic loop between TM6 and TM7 are disordered, as in most other LacY crystal structures. C-terminal residues 411–417 in chain A and 412–417 in chain B were also poorly ordered and are not included in the structure. In each complex, a molecule of TDG is observed in the substrate-binding site and Nb9043 is bound to the periplasmic surface of the C-terminal domain. In both complexes two n-Nonyl-β-D-glucopyranoside (β-NG) molecules are bound near the periplasmic vestibules, one at the interface with the Nb and one adjacent to the substrate binding site (S2 Fig). There are also five molecules of β-NG bound to outer surfaces of the proteins.

The conformation of LacY$_{WW}$ in the LacY$_{WW}$/TDG/Nb9043 complex is very similar to its conformation in the LacY$_{WW}$/TDG complex with no Nb bound (PDB ID 4OAA)[7]. The rmsd between 382 aligned α-carbon atoms of LacY$_{WW}$ in the two structures is 0.9 Å (Fig 1). In both complexes, the cytoplasmic side of LacY$_{WW}$ is closed and the periplasmic vestibule is partially occluded. Conformational differences are limited to the periplasmic loops and the periplasmic ends of the TMs, which are moved by 1-2Å.

The crystal structure of our previously determined LacY$_{WW}$/NPG/Nb9047 (PDB ID 6C9W) shows LacY$_{WW}$ in the same conformation as in LacY$_{WW}$/TDG/Nb9043; the rmsd between 387 aligned Cα atoms of LacY$_{WW}$ in the Nb9047 and Nb9043 complexes is 0.6 Å, close to the expected noise level at that resolution (Fig 2a)[20, 21]. Differences between LacY$_{WW}$ conformations in the two complexes are limited to periplasmic loops in direct contact with the Nbs. In contrast, LacY$_{WW}$ has a more open periplasmic vestibule in an X-ray structure of apo-LacY$_{WW}$/Nb9039 (Fig 2b)[19]. Here, the periplasmic ends of the TMs are shifted outward by ~2 to 3Å with respect to the TMs in the other two Nb complexes of LacY$_{WW}$. Thus, while Nb9043, Nb9047 and Nb9039 all bind to the periplasmic side of outward-open LacY$_{WW}$, binding of substrate to the Nb9043 and Nb9047 complexes stabilizes a more occluded periplasmic vestibule in LacY$_{WW}$.

## The LacY$_{WW}$/Nb9043 interface

As in previous Nb/LacY$_{WW}$ complex structures, the CDRs of Nb9043 interact with the ends of periplasmic helices and connecting loops in the C-terminal domain of LacY$_{WW}$[19, 20], but Nb9043 is shifted ~7Å toward the N-terminal domain in the open periplasmic cavity of LacY$_{WW}$ relative to the Nbs in previous structures (Fig 2). The new binding mode may result from the much shorter CDR3 in Nb9043 (8 residues shorter than the CDR3 of Nb9047), which doesn't extend as far into the periplasmic vestibule as the CDR3s of Nb9047 or Nb9039. In each complex of the asymmetric unit of LacY$_{WW}$/TDG/Nb9043, a molecule of the detergent β-NG from the protein solution used for crystallization binds adjacent to CDR3, with the glucose end facing the vestibule and partially overlapping the CDR3 binding site in LacY$_{WW}$/NPG/Nb9047. The β-NG mediates hydrogen bonds between the Nb and LacY$_{WW}$ and may stabilize the conformation of the complex in the crystal, but is not physiological.

Specific interactions between Nb9043 and LacY$_{WW}$ are very different from contacts observed in the previous Nb complexes because of the shift in position of Nb9043 (Fig 3). Protein interface analysis [22] shows that the LacY$_{WW}$-Nb9043 interface is 853Å$^2$ and the average solvation ΔG$^i$ for the A-C and B-D interfaces is -2 kcal/mol. Based on distances between protein donors and acceptors PISA identifies 10 hydrogen bonds and 4 salt bridges in the A-C interface and 13 hydrogen bonds and 4 salt bridges in the B-D interface. Inspection of the

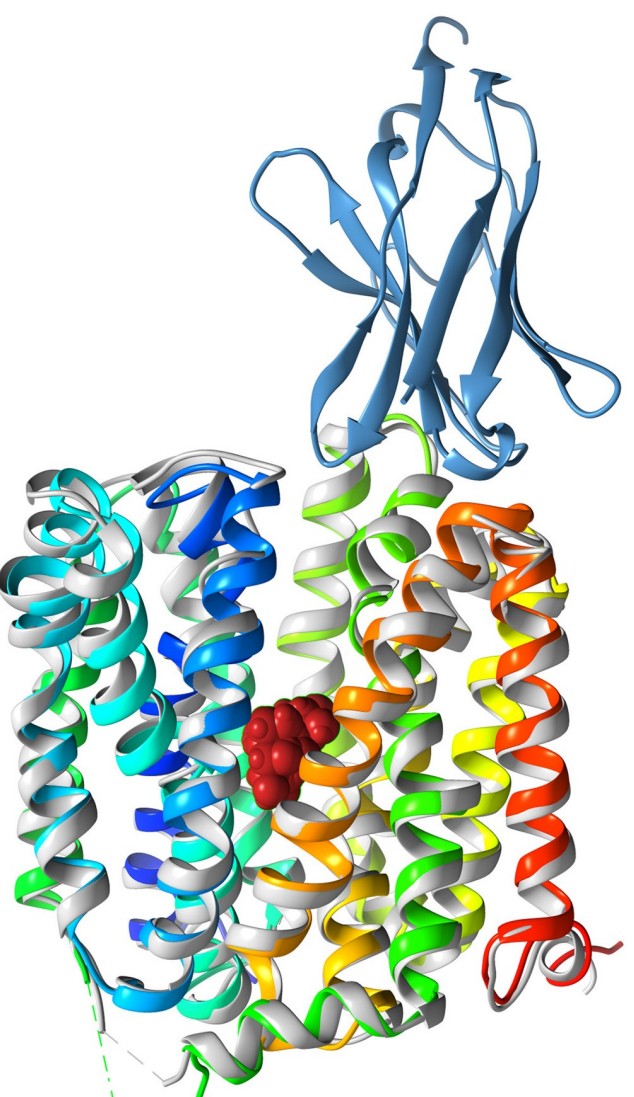

**Fig 1. Comparison of TDG-bound LacYww with and without bound Nb.** Cartoon drawing of LacY$_{WW}$/TDG/
Nb9043 (rainbow colored from blue at the N-terminus to red at the C-terminus) is superimposed on the cartoon of
LacY$_{WW}$/TDG (light gray). Differences between LacY$_{WW}$ in the two complexes are small and limited to the
periplasmic ends of transmembrane helices and connecting loops. Nb9043 is shown as blue ribbons and TDG is shown
as red space-filling spheres.

geometry of these interactions identifies 14 hydrogen bonds between Nb9043 and LacY$_{WW}$
that are conserved in the two complexes of the asymmetric unit. The few non-conserved inter-
actions and non-optimal hydrogen bond distances involve less well-ordered parts of the struc-
ture and reflect uncertainty in the conformation of some side chains. In one of the complexes
an ordered water molecule mediates hydrogen bonds between Nb9043 and LacY$_{WW}$. All likely
hydrogen bonds, both direct and small-molecule mediated, are listed in Table 1.

The LacY$_{WW}$ interface with Nb9047 is similar to the interface with Nb9043, comprising an
area 820Å$^2$, 13 hydrogen bonds, and solvation $\Delta G^i$ = -4.2 kcal/mol[22]. The interface with
Nb9039 is larger, 999Å$^2$, with 15 possible hydrogen bonds and solvation $\Delta G^i$ = -8.8 kcal/mol

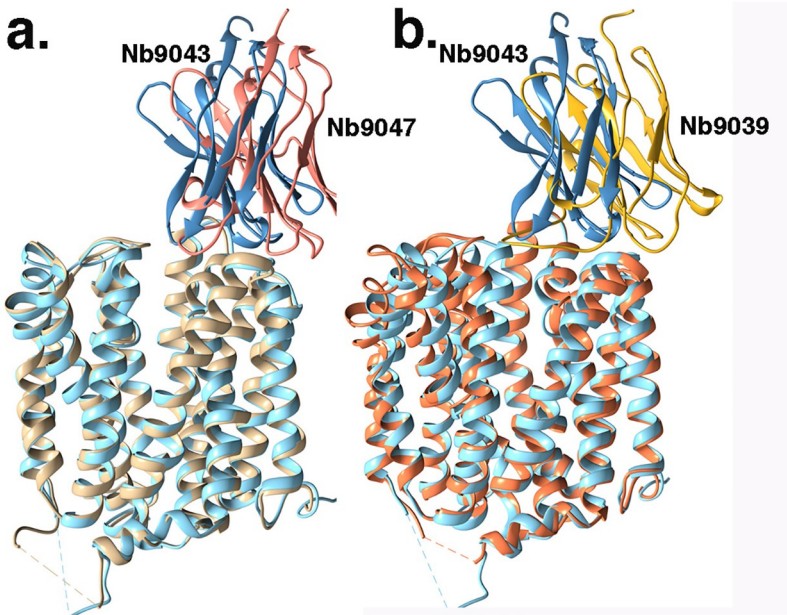

**Fig 2. Structural comparison of the Nb/LacYww complexes with or without bound galactoside.** a. LacY$_{WW}$/TDG/
Nb9043 is superimposed on LacY$_{WW}$/NPG/Nb9047 to illustrate strong conservation of the LacY$_{WW}$ conformation in
the two complexes. The Nbs use CDR3 for binding to the vestibule and are in the same general orientation though
shifted ~7Å. The structures are depicted as ribbons. LacY$_{WW}$ and bound Nb9043 are light blue/slate blue while
LacY$_{WW}$ and bound Nb9047 are tan and salmon, respectively. b. Superposition of LacY$_{WW}$/TDG/Nb9043 with
LacY$_{WW}$/Nb9039 shows more extensive shifts in TMs between the two complexes, particularly in the N-terminal
domain. LacY$_{WW}$ in the LacY$_{WW}$/Nb9039 complex is shown as brown ribbons and Nb9039 is drawn as gold ribbons.
LacY$_{WW}$/TDG/Nb9043 is colored blue as in a.

[22]. The larger interface with Nb9039 results from the larger CDR3 loop in contact with
LacY$_{WW}$.

A distinctive feature of the previously reported LacY$_{WW}$/NPG/Nb9047 complex (PDB ID
6C9W) is a contact area of the CDR3 loop, which forms hydrogen bonds with Lys42 (at the
periplasmic end of TM2) of LacY$_{WW}$ and also with Asn245 in LacY$_{WW}$ TM7[20]. These inter-
actions create a bridge between the N- and C-terminal domains of LacY$_{WW}$ across the peri-
plasmic vestibule. Nb9043, with a much smaller CDR3 does not form analogous hydrogen
bonds.

The only interactions between Nb9043 and the N-terminal domain are a hydrogen bond
from His35 Nε2 in TM1 donated to Asp103 Oᵟ2 in CDR3 (Table 1), and a close contact
between Asp36 (LacYww) and Asp103 (CDR3) side chain carboxyl groups. These interactions
position CDR3 directly over the periplasmic vestibule such that it partially blocks access to the
substrate-binding site. Asp36 is predicted to have an elevated pKa of 5, thus could make a
direct or water-mediated hydrogen bond with Asp103. There are four other direct hydrogen
bonds between LacY$_{WW}$ and CDR3: the guanidinium group of Arg99 in CDR3 donates hydro-
gen bonds to the backbone carbonyls of Thr248, Phe251 and Ala252 in loop 7–8 (Fig 3d).

CDR1 interacts exclusively with loop 11–12 of LacY$_{WW}$ (Fig 3b). The guanidinium group
of Arg27 donates a hydrogen bond to Gln374 Oε2 of LacY$_{WW}$ loop 11–12. The backbone
amide of Phe378 in loop 11–12 donates a hydrogen bond to Asn31 Oᵟ1 of CDR1 and the back-
bone carbonyl of Tyr373 in loop 11–12 accepts a hydrogen bond from Asn31 Nᵟ2.

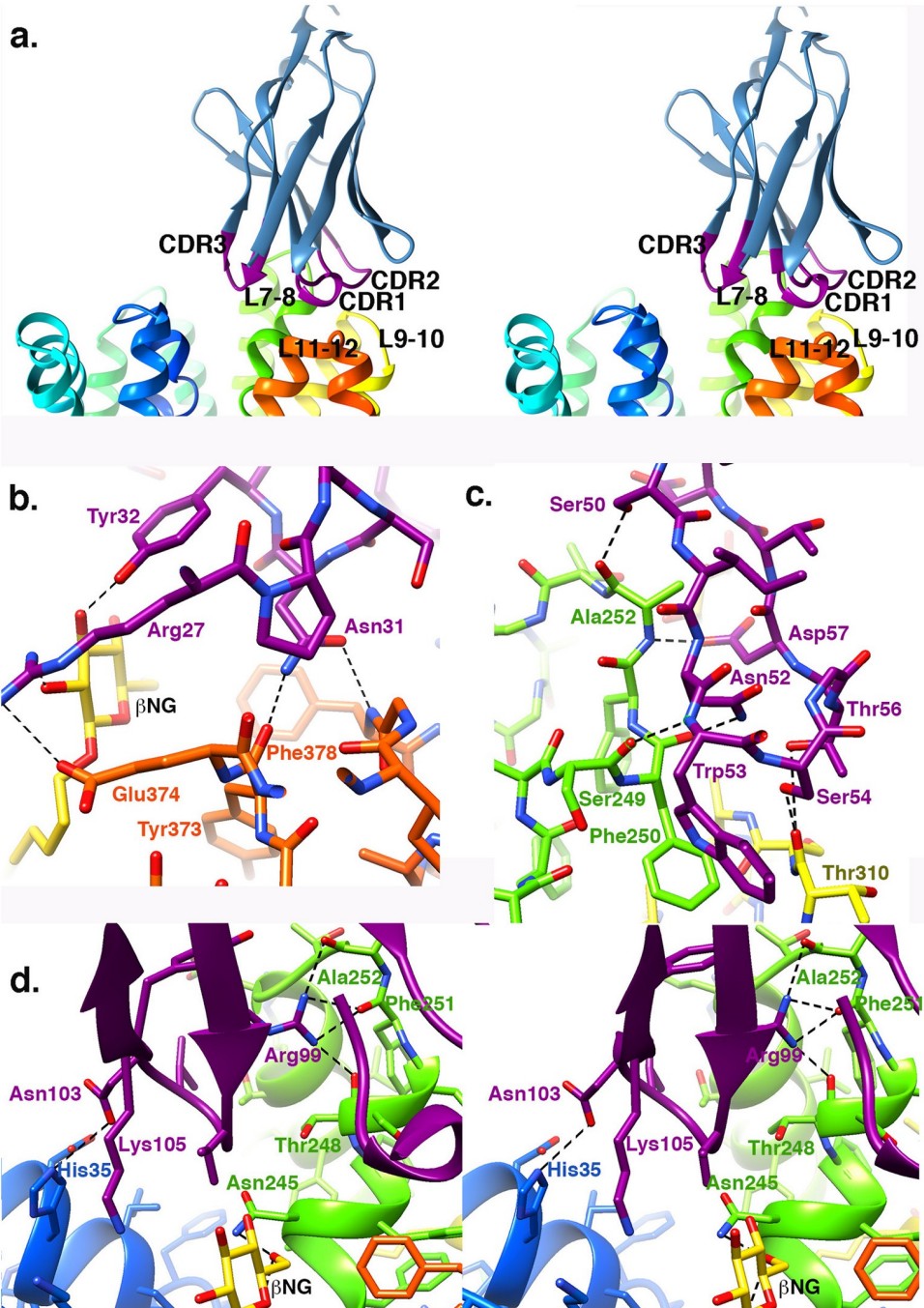

**Fig 3. Specific interactions between Nb9043 and LacY_WW.** a. Crossed-eyes stereo ribbon depiction of the interface between Nb9043 and LacY_WW. Nb9043 is mostly blue with the CDRs colored magenta and labeled. LacY_WW is shown in rainbow colors ranging from blue at the N-terminus to red at the C-terminus. Loops that interact with the CDRs are labeled. b. Plot of the hydrogen bonds between CDR1 (magenta carbons, red oxygens, blue nitrogens) and the loop between TM11 and TM12 of LacY_WW (orange carbons, red oxygens, blue nitrogens) and β-NG (gold carbons, red oxygens). Hydrogen bonds are shown with black dashed lines. c. Plot of the hydrogen bonds between CDR2 (magenta carbons, red oxygens, blue nitrogens) and the loop between TM7 and TM8 of LacY_WW (green carbons, red oxygens, blue nitrogens) and the loop between TM9 and TM10 (yellow carbons, red oxygens, blue nitrogens). Hydrogen bonds are shown with black dashed lines. d. Crossed-eyes stereo ribbon plot of CDR3, CDR1, and the TMs surrounding the periplasmic vestibule. Side chains for LacY_WW and CDR3, and the backbone atoms for Thr248, Phe251 and Ala 252 are drawn as sticks colored by atom type as in (3b) with the TM1 carbons in blue, TM7-TM8 carbons in green, CDR3 carbons in magenta and β-NG carbons in gold.

**Table 1. LacY$_{WW}$-Nb9043 interface hydrogen bonds.**

| LacY$_{WW}$ A-Nb C interface | | | LacY$_{WW}$ B-Nb D interface | | |
|---|---|---|---|---|---|
| LacY$_{WW}$ A | Nb C | Dist. (Å) | LacY$_{WW}$ B | Nb D | Dist. (Å) |
| **CDR1 interactions** | | | | | |
| E374 Oε2 | R27 Nη1 | 3.0 | E374 Oε2 | R27 Nη1 | 2.2 |
| F378 N | N31 Oδ1 | 2.7 | F378 N | N31 Oδ1 | 2.7 |
| Y373 O | N31 Nδ2 | 2.7 | Y373 O | N31 Nδ2 | 2.6 |
| **CDR2 interactions** | | | | | |
| A252 O | S50 Oγ | 2.5 | A252 O | S50 Oγ | 3.7 |
| F250 O | N52 Nδ2 | 2.5 | F250 O | N52 Nδ2 | 3.4 |
| S249 O | W53 N | 3.2 | S249 O | W53 N | 3.2 |
| T310 O | S54 Oγ | 2.6 | T310 O | S54 Oγ | 2.9 |
| T310 O | T56 Oγ1 | 3.6 | T310 O | T56 Oγ1 | 2.9 |
| A252 N | D57 Oδ1 | 3.1 | A252 N | D57 Oδ1 | 3.0 |
| **CDR3 interactions** | | | | | |
| F251 O | R99 Nη1 | 3.1 | F251O | R99 Nη1 | 3.1 |
| A252 O | R99 Nη1 | 3.0 | A252 O | R99 Nη1 | 2.7 |
| F251 O | R99 Nη2 | 2.8 | F251 O | R99 Nη2 | 2.8 |
| T248 O | R99 Nη2 | 2.8 | T248 O | R99 Nη2 | 2.8 |
| H35 Nε2 | D103 Oδ2 | 3.3 | H35 Nε2 | D103 Oδ2 | 3.8 |
| **Water-mediated interactions** | | | | | |
| T253 O | HOH 801 | 2.4 | | | |
| HOH 801 | Y104 Oη | 2.8 | | | |

Of the three CDR regions, CDR2 makes the most hydrogen bonds to LacY$_{WW}$ (Fig 3c). Side chains of five residues in CDR2 donate hydrogen bonds to backbone carbonyls in loops 7–8 and 9–10 of LacY$_{WW}$. The backbone carbonyl of Ser249 in LacY$_{WW}$ accepts a hydrogen bond from the backbone amide of Trp53 of CDR2.

## Galactoside-binding site

The sugar-binding site of LacY, centered between N- and C-terminal domains in the middle of the transmembrane region, is highly conserved between LacY$_{WW}$/TDG complexes with and without Nb9043 bound. While the periplasmic ends of the TM helices shift slightly toward the vestibule when Nb9043 is bound, the middle and cytoplasmic regions of the TMs closely align (Fig 1). There is well-resolved density for TDG in the substrate-binding sites of both LacY$_{WW}$/TDG/Nb9043 complexes of the asymmetric unit of the crystal structure (Fig 4). The TDG is in the same conformation as TDG in the previously determined LacY$_{WW}$/TDG complex (PDB ID 4OAA) and interacts with residues shown by mutagenesis, biochemistry and crystallography to be important for galactopyranosyl specificity, including Glu269, Arg144, Glu126, Trp151, and His322[7]. Table 2 lists hydrogen bond interactions between TDG and LacY$_{WW}$, also shown in Fig 4. A molecule of β-NG is bound adjacent to the substrate binding site in each complex, with its glucopyranosyl ring packed in a parallel fashion against one of the galactopyranosyl rings of TDG (S2 Fig). The glucopyranosyl ring substituents make potential hydrogen bonds with Asn119, Tyr350 and Cys355.

## Discussion

Nbs generated against stabilized periplasmic-open mutant LacY$_{WW}$[6], bind tightly and specifically to the periplasmic side of WT LacY, blocking lactose/H$^+$ symport across the membrane,

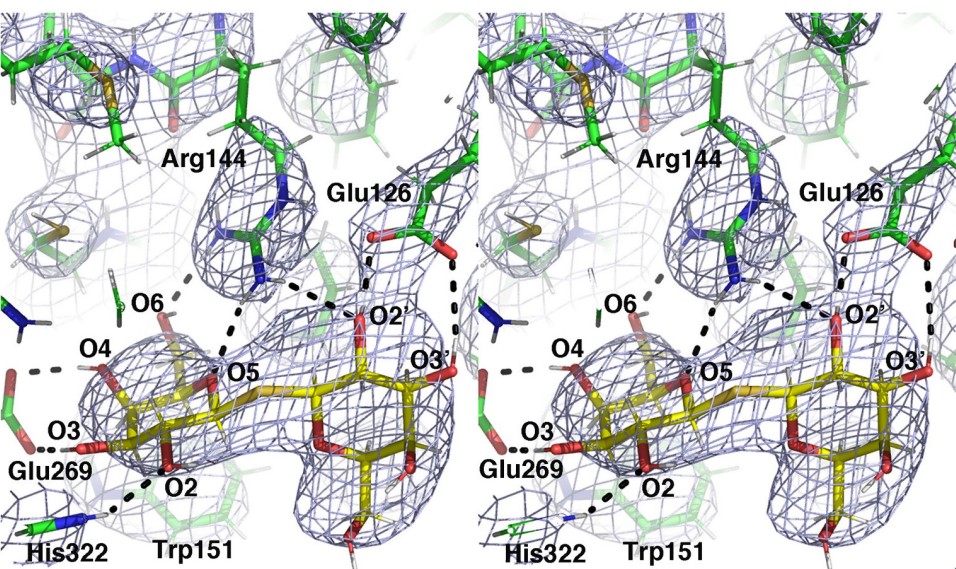

**Fig 4. LacY$_{WW}$-TDG interactions in LacY$_{WW}$/TDG/Nb9043.** Crossed-eyes stereo plot of TDG (sticks with gold carbons, red oxygens and white hydrogens) and surrounding LacY$_{WW}$ residues (sticks with green carbons, red oxygens, blue nitrogens, yellow sulfurs, and white hydrogens) overlaid with the 2mFo-DFc composite omit map contoured at 1σ. Hydrogen bonds to LacY$_{WW}$ residues are shown with dashed lines.

but increasing accessibility of the sugar-binding site from the periplasmic surface[17]. Crystal structures of two of these Nbs, Nb9039 and Nb9047, in complex with LacY$_{WW}$ were determined previously[19, 20]. We here report the 2.8Å structure of LacY$_{WW}$ complexed with a third nanobody, Nb9043 and containing TDG, a high-affinity lactose analog, bound at the sugar-binding site. This structure has the highest resolution of the LacY-Nb structures obtained so far and clearly defines most of the side chains, main chains and hydrogen bonded interactions at the LacY$_{WW}$/Nb interface. In all three of the LacY$_{WW}$/Nb complexes, the CDR3 loops, which differ from each other in length and sequence, bind to the periplasmic side forming a bridge between N- and C-terminal six-helix bundles which would limit the conformational change in WT LacY by trapping an outward-open state. Thus, this periplasmic binding site seems to be an immunological hotspot on LacY and is consistent with the abilities of the

**Table 2. LacY$_{WW}$−TDG hydrogen bonds (two complexes of the asymmetric unit are marked A and B).**

| TDG A–LacY$_{WW}$ A hydrogen bonds | | | TDG B–LacY$_{WW}$ B hydrogen bonds | | |
|---|---|---|---|---|---|
| TDG A | LacY$_{WW}$ A | Dist (Å) | TDG B | LacY$_{WW}$ B | Dist. (Å) |
| O2 (A) | H322 Nε2 | 2.8 | O2 (A) | H322 Nε2 | 2.6 |
| O3 (D) | E269 Oε2 | 2.4 | O3 (D) | E269 Oε2 | 2.4 |
| O4 (D) | E269 Oε1 | 2.7 | O4 (D) | E269 Oε1 | 2.7 |
| O5 (A) | R144 Nη1 | 2.9 | O5 (D) | R144 Nη1 | 2.8 |
| O6 (A) | R144 Nη2 | 3.1 | O6 (A) | R144 Nη2 | 3.1 |
| O2' (A) | R144 Nη1 | 3.4 | O2' (A) | R144 Nη1 | 3.3 |
| O2' (D) | E126 Oε2 | 2.4 | O2' (D) | E126 Oε2 | 2.2 |
| O3' (D) | E126 Oε1 | 3.0 | O3' (D) | E126 Oε1 | 2.9 |

Nbs to block lactose transport in WT LacY and increase accessibility for sugar binding from the periplasmic side[17].

Even though Nb9043 has a shorter CDR3 loop than Nb9047 and binds to LacY$_{WW}$ at a shifted position, the LacY$_{WW}$ conformations in complexes with the two nanobodies are almost the same. Distance-dependent Trp quenching experiments, in which Trp and Cys pairs were introduced into WT LacY so that a fluorescent probe attached to Cys is quenched or unquenched in response to LacY conformational changes, support the similarity of the LacY conformations in the Nb9043 and Nb9047 complexes. Thus, in engineered bimane-labeled LacY fluorescence quenching/unquenching was approximately equally effective in complexes formed with Nb9043 and Nb9047[18].

A factor contributing to the nearly identical LacY$_{WW}$ conformations in the Nb9043 and Nb9047 complexes is the presence of bound sugar in both complexes. Sugar binding to Nb/LacY complexes has been shown to make additional conformational changes in LacY, enhancing quenching/unquenching levels to similar values for several Nb-LacY complexes[18]. Thus, Nb9047 and Nb9043 complexes with LacY$_{WW}$ may differ more in conformation in the absence of bound substrate, but sugar-binding is a factor in determining the conformation seen in the crystal structures. It is perhaps for this reason we were unable to crystallize Nb9047 and Nb9043 complexes in the absence of galactoside. Correspondingly Nb9039 complex did not bind substrate.

The conformation of LacY$_{WW}$ is much more periplasmic-open in apo-LacY$_{WW}$/Nb9039. The LacY$_{WW}$ conformational differences between LacY$_{WW}$/Nb9039 and the other two Nb complexes occur as shifts of the periplasmic ends of the TMs mainly in TM1 –TM4. The structures of the substrate-binding sites in the center of LacY$_{WW}$ are conserved in the three Nb complexes, with hydrogen bonding groups ideally oriented for binding a galactosyl moiety, thus it is unlikely that binding of sugar to apo-LacY$_{WW}$/Nb9039 would induce closure of the periplasmic vestibule to a conformation more similar to the sugar-bound LacY$_{WW}$-Nb complexes. Rather, the more open LacY$_{WW}$ conformation has been selected naturally during generation of Nb9039 with its larger CDR3 loop that optimizes interactions with antigen. In particular, in apo-LacY$_{WW}$/Nb9039 a connection between the N- and the C-terminal domains is mediated by hydrogen bonds of CDR3 formed with the side chains of both Lys42 and Asn245 of LacY. In LacY$_{WW}$/NPG/Nb9047, Asn245 is linked to the backbone rather than the side chain of Lys42 by Nb hydrogen bonds, stabilizing a more closed vestibule than in the Nb9039 complex. The small CDR3 of Nb9043 does not bridge Lys42 and Asn245 by hydrogen bonds. However, Asp103 O™2 in CDR3 of Nb9043 accepts a hydrogen bond from His35 Nε2 in the N-terminal domain of LacY$_{WW}$ and this interaction may be important for stabilizing a particular periplasmic-open conformer of WT LacY.

The rate of opening of the periplasmic cavity is rate-limiting for sugar binding[23]. Smirnova et al. reported dramatically elevated "on-rates" ($k_{on}$) for galactoside binding to WT LacY/Nb complexes (increases from 0.2 $\mu M^{-1}s^{-1}$ for WT LacY to 4.4, 6.9, and 2.9$\mu M^{-1}s^{-1}$ for WT LacY complexes with Nb9039, Nb9047 and Nb9043, respectively, S3A Fig), which indicate greater accessibility of the sugar-binding site in periplasmic-open conformations trapped by each of the three Nbs (S1 Table)[17]. They found that mutation of WT LacY to LacYww did not restrict access to the sugar-binding site ($k_{on}$ = 5.7 $\mu M^{-1}s^{-1}$), nor did binding of Nb9039 or Nb9047: $k_{on}$ for the LacY$_{WW}$/Nb9039 or LacY$_{WW}$/Nb9047 complexes were practically the same as $k_{on}$ for unbound LacY$_{WW}$ (S3B Fig and S1 Table)[17]. However, they found $k_{on}$ for LacYww/Nb9043 was much lower (2.2 $\mu M^{-1}s^{-1}$) and exhibited a similarly low $k_{on}$ (2.9 $\mu M^{-1}s^{-1}$) for the Nb9043 complex with WT LacY. This supports that the Nb binds to WT LacY and then biases the solution conformation to the periplasmic-open form. Moreover, rates of dissociation ($k_{off}$) of bound sugar from LacY/Nb complexes decreased with respect to protein not bound to

Nb only for complexes of Nb9043 with either WT LacY or LacYww (S4 Fig and S1 Table)[17]. Much slower $k_{off}$ values (5–6 times) indicate that sugar dissociation from the binding site is sterically more difficult when Nb9043 is bound to LacY compared to when Nb9039 or Nb9047 are bound. The similarity of sugar-binding kinetic parameters measured for WT LacY/Nb9043 and LacY$_{WW}$/Nb9043 most likely reflects a structural resemblance of LacY conformers and their interactions with Nb9043 in the two complexes. In other words, the outward-open conformer of WT LacY trapped by bound Nb9043 has essentially the same overall structure as LacY$_{WW}$ in complex with Nb9043. This emphasizes the important conclusion that the Nbs made against biased conformations of LacY, do indeed stabilize WT LacY in the conformation against which they were made.

The lower $k_{on}$ and $k_{off}$ rates (but similar $K_d$ values) for LacY or LacYww complexes with Nb9043 compared to complexes with Nb9039 and Nb9047 are consistent with the partial occlusion of the periplasmic cavity by CDR3 in the structure of LacY$_{WW}$/Nb9043. Although in LacY$_{WW}$/Nb9047 LacY$_{WW}$ has the same conformation as in LacY$_{WW}$/Nb9043, and the Nb9047 CDR3 loop also binds in the mouth of the periplasmic vestibule, the periplasmic opening leading to the substrate-binding site is less occluded. The greater accessibility of the periplasmic cavity in LacY$_{WW}$/Nb9047 results in part from the high Gly content of CDR3 of Nb9047. The four Gly residues in CDR3 of Nb9047 make it less bulky and more flexible, thus allowing it to bind more closely to the vestibule wall, rather than jutting into the periplasmic vestibule.

## Conclusions

Nanobody Nb9043 made against LacY$_{WW}$, a double mutant constrained to be in a periplasmic-open conformation, binds to, and stabilizes the periplasmic-open conformations of wild type LacY as shown by a dramatic increase in $k_{on}$ for sugar binding to the WT LacY/Nb complex. A 2.8Å crystal structure of LacY$_{WW}$/Nb9043 shows that it binds in the same general region of LacY as two other Nbs that were co-crystallized with LacY$_{WW}$, but its position is shifted with respect to the other Nbs. In all three complexes, the Nbs are bound primarily to the C-terminal domain of LacY$_{WW}$ and in this case, the Nb CDR3 binds in the periplasmic vestibule. However, the detailed interactions of Nb9043 with the symporter are completely different than those of the other Nbs because of its shifted position. CDR3 of Nb9043 partially occludes the periplasmic cavity, blocking substrate access, and leading to much lower substrate $k_{on}$ and $k_{off}$ than for unbound LacY$_{WW}$. The fact that we were unable to crystallize a WT LacY/Nb9043 complex indicates the Nb probably does not bind strongly enough to stabilize a periplasmic-open conformation of WT-LacY for crystallization; however, similar binding kinetics for the Nb9043 complex with WT LacY indicate the LacY$_{WW}$ and WT LacY complexes with Nb9043 have similar structures, validating the high value of LacY$_{WW}$/Nb crystal structures for mapping outward-open LacY conformations.

## Materials and methods

### Materials

TDG was obtained from Carbosynth Limited and buffers were from Sigma-Aldrich. Talon superflow resin was from BD Clontech. Dodecyl-β-D-maltopyranoside (DDM) and n-nonyl-β-D-glucoside (β-NG) were from Affymetrix. All other materials were of reagent grade and obtained from commercial sources.

Preparation of Nb9043 was previously described in detail[7]. Inducible periplasmic expression of His-tagged Nb in *E. coli* yields >95% pure protein using immobilized metal ion affinity

chromatography (Talon resin). Purified Nb9043 (20 mg/ml) in 100 mM potassium phosphate (KP$_i$, pH 7.5) was frozen in liquid nitrogen and stored at -80˚C prior to use.

### Growth, expression, and purification

Plasmid pT7-5 encoding LacY$_{WW}$ with a 6-His-tag was expressed in *E. coli* C41. A detailed procedure for protein purification has been described[8]. Briefly, membranes were prepared and solubilized in 2% DDM, and LacY$_{WW}$ was purified on a Talon column. Protein was eluted in 20 mM HEPES, 0.2% β-NG, 200 mM imidazole buffer (pH 6.5). Purified Nb9043 was added to LacYww in a 3:1 molar ratio (400 ul, 10 mg/ml LacY$_{WW}$ + 150 μl 7 mg/ml Nb9043 + 3 μl *E. coli* polar Phospholipid 40 mg/ml) and subjected to size-exclusion chromatography. A homogeneous peak containing the LacY$_{WW}$/Nb9043 complex was eluted in 20 mM HEPES /0.2% β-NG (pH 6.5) and used for crystallization.

### Crystallization, data collection, and structure determination

TDG was added (5 mM, final concentration) to a protein solution (10 mg/ml in 20 mM HEPES, 0.2% β-NG, pH 6.5) prior to crystallization trials. Crystallization screens were performed using the hanging-drop vapor diffusion method on a Mosquito Crystal Robot (TTP Labtech) in a 96-well plate with a drop ratio of 200 nl well solution to 200 nl protein. Crystals appeared in 24 h and grew to the size of ~100 μm in a week. Crystals were harvested and screened for diffraction. Initially they diffracted to 6–7 Å. The drop containing crystals was sealed after harvesting the crystals and the plate was stored at 20˚C. Before throwing away the plate after 2 years, the crystals were again harvested and taken to the synchrotron, where the crystals then diffracted to 2.8 Å. Although crystals appeared in wide range of PEG concentrations and at different pHs with different buffers, the best diffracting crystals grew in Memgold screen well F12 (0.07M sodium chloride. 0.05M sodium citrate pH 4.5, 22% v/v PEG 400). No cryoprotectant was added prior to data collection.

Diffraction data were collected at the Lawrence Berkeley National Laboratory Advanced Light Source Beamline 8.3.1, at −170 ˚C at a wavelength of 1.115 Å. The best crystals diffracted to 2.8 Å resolution. Data were indexed with HKL2000[24] and processed with XDS[25]. The structure was determined by molecular replacement using LacY$_{WW}$ alone from LacY$_{WW}$/Nb9047/NPG (PDB ID 6C9W) as the search model. To build Nb9043, the available structure of camelid Nb9047 (PDB ID code 6C9W) was manually aligned with the 2Fo-Fc density, side chains and CDRs from Nb9043 were fitted and refined. The program COOT was used for density fitting[26], and refinement was carried out using PHENIX[27] and Refmac from the CCP4 suite[28]. Hydrogen atoms were included in riding positions. Refinement proceeded with manual rebuilding with the assistance of sigmaA-weighted 2Fo-Fc, Fo-Fc, and complete omit maps, alternating with maximum-likelihood based energy minimization and isotropic B-factor refinement. The resulting model improved R$_{work}$ and R$_{free}$ to 0.239 and 0.275, respectively. The MolProbity server[29] was employed for structural assessment and validation. The final round of refinement improved the stereochemistry and geometry of the structure, with 96.0% of residues in the favored region of the Ramachandran plot and 3.9% in the allowed region. The data and refinement statistics are shown in S2 Table. Figs 1–3, S1 and S2 Tables were made with the UCSF Chimera Package[30]. Fig 4 was made with PyMOL (The PyMOL Molecular Graphics System, Version 1.4.1, Schrodinger, LLC).

## Supporting information

**S1 Fig. Packing in the LacY$_{WW}$/TDG/Nb9043 crystal.** An asymmetric unit of the crystal is shown in crossed-eyes stereo with tan (A-C complex) and brown (B-D complex) ribbons, with

chains labeled in bold letters. Two of the symmetry mates in the unit cell that interact with this asymmetric unit are shown as grey and light blue ribbons, with interacting chains labeled with oblique letters.
(TIF)

**S2 Fig. β-NG molecules co-crystallized with LacY$_{WW}$/TDG/Nb9043.** Cartoon version of one LacY$_{WW}$/TDG/Nb9043 complex (chains A, C) showing β-NG and TDG molecules drawn as sticks and colored by atom type (red oxygens, blue nitrogens, yellow sulfur and gold carbons (for β-NG) or blue carbons (for TDG). LacY$_{WW}$ is colored green and Nb9043 is purple. Two β-NG molecules bound in the periplasmic vestibules, one at the Nb9043 interface and one near the substrate-binding site, are boxed and their interactions with LacY$_{WW}$ are shown on the right, with hydrogen bonds drawn with dashed lines. Regions of a composite omit 2mFo-DFc map surrounding the β-NGs are overlaid with the plots on the right and shown as grey mesh.
(TIF)

**S3 Fig. Effect of Nbs on kinetics of NPG binding to WT LacY (A) or LacY$_{ww}$ (B).** Galacto-side binding rates reported by Smirnova et al.[17] were measured by stopped-flow as change in Trp fluorescence utilizing FRET from Trp151 of LacY to bound NPG. Concentration dependencies of the binding rates observed ($k_{obs}$) were measured before, or after preincubation of LacY with 1.5-fold excess of Nbs. Data shown in blue, pink, and red correspond to LacY complexes with Nb9039, Nb9047, and Nb9043, respectively. Binding rates in the absence of Nb are shown in green and black for WT LacY (A), and LacYww (B), respectively. The $k_{on}$ values were calculated from the slopes of linear fits and presented in S1 Table.
(TIF)

**S4 Fig. NPG dissociation from LacY/NPG complexes reported by Smirnova et al. [17].** The $k_{off}$ values were measured by stopped-flow as Trp fluorescence increase resulting from dis-placement of bound NPG (acceptor of FRET from Trp151) by excess of TDG. Single exponen-tial fits (black lines) of stopped-flow traces are shown for WT LacY (A) and LacYww (B) in blue, pink, and red for LacY complexes with Nb9039, Nb9047, and Nb9043, respectively. LacY (0.5 μM) preincubated with 1.5-fold excess of Nbs was mixed first with NPG, and after 10 min with TDG. Final concentrations of NPG and TDG were 0.1 and 10 mM, respectively. The $k_{off}$ values are presented in S1 Table.
(TIF)

**S1 Table. Kinetic parameters of galactoside binding to the complexes of LacY, and LacY with Nbs [17].**
(RTF)

**S2 Table. Data collection and refinement statistics for LacY$_{WW}$/TDG/Nb9043.**
(RTF)

## Author Contributions

**Conceptualization:** Hemant Kumar, H. Ronald Kaback, Robert M. Stroud.

**Formal analysis:** Hemant Kumar, Irina Smirnova, Vladimir Kasho, H. Ronald Kaback, Robert M. Stroud.

**Funding acquisition:** H. Ronald Kaback, Robert M. Stroud.

**Investigation:** Hemant Kumar, Irina Smirnova, Vladimir Kasho, Els Pardon, Jan Steyaert.

**Resources:** Els Pardon, Jan Steyaert.

**Supervision:** H. Ronald Kaback, Robert M. Stroud.

**Writing – original draft:** Irina Smirnova, Vladimir Kasho, H. Ronald Kaback, Robert M. Stroud.

**Writing – review & editing:** Hemant Kumar, Irina Smirnova, Vladimir Kasho, H. Ronald Kaback, Robert M. Stroud.

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
