## [Decision Letter · Decision Letter 0]

25 Mar 2020

PONE-D-20-03505

Nanobodies against a locked, altered state of a transporter show remarkable diversity in binding modes: Structure of a lactose permease-nanobody complex

PLOS ONE

Dear Professor Stroud,

Thank you for submitting your manuscript to PLOS ONE. Your manuscript was reviewed by two expert reviewers whose comments are added at the end of this message. After careful consideration, we feel that your manuscript has merit but does not yet fully meet PLOS ONE’s publication criteria as it currently stands. In addition to the general and specific comments of both reviewers, both reviewers suggest that a change in the title of your manuscript is required as the reported state of the transporter is similar to one previously published. 

We invite you to submit a revised version of the manuscript that addresses the points raised during the review process.

We would appreciate receiving your revised manuscript by May 09 2020 11:59PM. To enhance the reproducibility of your results, we recommend that if applicable you deposit your laboratory protocols in protocols.io, where a protocol can be assigned its own identifier (DOI) such that it can be cited independently in the future. For instructions see: http://journals.plos.org/plosone/s/submission-guidelines#loc-laboratory-protocols

We look forward to receiving your revised manuscript.

Kind regards,

Hendrik W. van Veen, PhD

Academic Editor

PLOS ONE

Journal Requirements:

Please ensure that your manuscript meets PLOS ONE's style requirements, including those for file naming. The PLOS ONE style templates can be found at http://www.plosone.org/attachments/PLOSOne_formatting_sample_main_body.pdf and http://www.plosone.org/attachments/PLOSOne_formatting_sample_title_authors_affiliations.pdf

Reviewers' comments:

Reviewer's Responses to Questions

**Comments to the Author**

1. Is the manuscript technically sound, and do the data support the conclusions?

Reviewer #1: Yes

Reviewer #2: Yes

2. Has the statistical analysis been performed appropriately and rigorously? 

Reviewer #1: Yes

Reviewer #2: N/A

3. Have the authors made all data underlying the findings in their manuscript fully available?

Reviewer #1: Yes

Reviewer #2: Yes

4. Is the manuscript presented in an intelligible fashion and written in standard English?

Reviewer #1: Yes

Reviewer #2: Yes

5. Review Comments to the Author

Reviewer #1: Dear Authors (and Editors)

As I am also involved in the field of membrane protein structure, I really understand how it is very hard to improve the resolution even only for 0.2Å (previous structure: 3.0Å -> this paper: 2.8Å). And in the crystallography, the 2.8Å resolution is what we called as “atomic resolution”. At last, the outward conformation of LacY has got into the atomic level. So, I may admire this work. The methods used in this paper to solve the structure are standard crystallography, similar to many other membrane protein structures. The approach appears to be valid and the data is of sufficient quality to support most of the conclusions in the paper.

But, unfortunately, the manuscript reads as an anticlimax.

Authors show the structure of the LacYww(G46W/G262W) in complex with Nanobody (Nb)#9043 and bound TDG. Previously, the same research group reported the same LacYww with different Nb #9047 and bound similar substrate, NPG. Before that, they also reported the same LacYww with bound TDG, but without any bound Nb. All these three structures have almost the same structures. Indeed, the structure shown in this paper is the highest among these structures, but unfortunately, there is no exciting “NEW” in understanding the molecular mechanism of this interesting transporter proteins, LacY.

Again, it is the highest “record” in the resolution of the outward conformation of LacY. But most of the structural aspects were discussed already in their previous papers.

Thus, it is remarkable as a record, but not very exciting in this field of science.

Minor comments:

Line 144, two “Nb9039”

Are they Nb9043?

Line 217 “antibodies“

The authors described as “antibodies” only once here. For consistency in the terminology, it should be “nanobodies” throughout the paper.

Line 226, “Nb9047”

Is this Nb9039?

Line 310, “A degree”

Use the proper font for angstrom.

Line 315, “sodium citrate 4.5, ”

Insert pH as “sodium citrate pH 4.5, ”

Reviewer #2: Kumar et al describes the crystal structure of LacYww (a double Trp mutant stabilized in its outward-facing state) in complex with nanobody Nb9043 and TDG in a periplasmic open orientation at 2.8Å resolution, which represents the highest resolution structure of this MFS transporter. At this resolution, the authors could accurately determine the hydrogen-bond network that bridges the N- and C-terminal domains. In previous studies, the authors reported the LacYww structures with two other nanobodies binding at the same general region, indicating that this region is an immune hotspot. In contrast to the other nanobodies, LacYww in complex with Nb9043 exhibits lower kon and koff rates for sugar binding, which can be explained by blocking the substrate accessibility by CDR loops of the nanobody. The high resolution, the different binding mode of the co-crystallized nanobody and hindrance of sugar binding as a consequence of nanobody binding justifies publication of this work.

Major points

1) The title is overly complicated, we suggest to simplify it.

2) Line 134 ff: we strongly suggest to analyze the nanobody/LacY interface using the PISA server (or a similar resource). This allows for an objective analysis, including also a calculation of the binding interface, numbers of hydrogen bonds and number of salt bridges. It would be also interesting to see a comparison with the previously characterized nanobodies using the PISA server.

3) Table S1, Fig. S2 and Fig. S3 show previously published data (Ref 17). It is perfectly fine to include these previously published data as supplementary Figures to have all relevant data complete. However, previous publication of the data needs to be stated clearly in the main text as well as in the respective Figure/Table legends!

Minor points

1) Line 80/81: Please comment on crystal packing of the two LacY/Nb complexes on the ASU.

2) Line 84-88: The authors describe in detail, where the two bound β-NG’s are located. It would be nice to see this in a Figure.

3) Along the same line: In Figure S1, the authors show the detailed interaction between β-NG and LacYww. However, it would be nice to see a larger picture to know where this interaction takes place and then as a zoom-in one could show what presently is Figure S1. Furthermore, to better distinguish β-NG from LacYww (carbons shown in green for both), the authors could use gold for the β-NG carbons (as in Figure 3d).

4) Line 83/84: please re-write the sentence “Also poorly ordered, C-terminal residues 412-417 in one molecule of LacYWW and 411-417 in the other were not fitted to density maps.” The sentence is currently difficult to grasp.

5) Line 97: remove “LacYWW/TDG/Nb9043 is” before “Differences between LacYww…”

6) Line 129-133: Although the second β-NG binding site is not physiological, it would be nice to see this in a figure (maybe in updated Figure S1? See comment above).

7) Line 144: In Figure 3 you show Nb9043 and not Nb9039.

8) Figure 3b: we suggest to label Phe378 and Tyr373.

9) Figure 3c: Numbering of the residues from TM7 and TM8 are confusing. It is hard to distinguish whether the numbers belong to the CDR or not (especially: Ala252).

10) Figure 3d: Not only CDR3 is shown, but also residues from CDR1. In the figure legend, only CDR3 residues are mentioned. But Tyr32 belongs to CDR1, correct? Thr248 is hidden behind Tyr32. It is not clear between which residues the hydrogen bond (dashed line) is. The one between Asn245 and Thr248 or between β-NG and Tyr32?

11) Line 168: It is hard to see the three hydrogen bonds in Fig. 3d (would it be possible to also label Phe251 and Ala252 in Figure 3d?).

12) Line 188-191: We suggest that the authors show how β-NG binds to the substrate binding site in a Figure.

13) Line 246: The authors write about an 10-20 fold increased on-rate, whereas in the introduction they wrote that the rate of sugar binding increases by 5-50 fold. Please clarify.

6. PLOS authors have the option to publish the peer review history of their article (what does this mean?). If published, this will include your full peer review and any attached files.

Reviewer #1: No

Reviewer #2: Yes: Markus Seeger and Lea Huber-Hürlimann

---

## [Author Response · Author response to Decision Letter 0]

21 Apr 2020

Reviewer comments have been addressed in the response_to_reviewers file

---

## [Editor Report · Decision Letter 1]

23 Apr 2020

Diversity in kinetics correlated with structure in nano body-stabilized LacY

PONE-D-20-03505R1

Dear Prof. Stroud,

We are pleased to inform you that your manuscript has been judged scientifically suitable for publication and will be formally accepted for publication once it complies with all outstanding technical requirements.

With kind regards,

Dr Hendrik van Veen

Academic Editor

PLOS ONE

---

## [Editor Report · Acceptance letter]

28 Apr 2020

PONE-D-20-03505R1 

Diversity in kinetics correlated with structure in nano body-stabilized LacY 

Dear Dr. Stroud:

I am pleased to inform you that your manuscript has been deemed suitable for publication in PLOS ONE. Congratulations! Your manuscript is now with our production department. 

With kind regards,

on behalf of

Dr Hendrik W. van Veen 

Academic Editor

PLOS ONE